# Genetic and Chemical Engineering of Phages for Controlling Multidrug-Resistant Bacteria

**DOI:** 10.3390/antibiotics10020202

**Published:** 2021-02-19

**Authors:** Dingming Guo, Jingchao Chen, Xueyang Zhao, Yanan Luo, Menglu Jin, Fenxia Fan, Chaiwoo Park, Xiaoman Yang, Chuqing Sun, Jin Yan, Weihua Chen, Zhi Liu

**Affiliations:** 1Department of Biotechnology, College of Life Science and Technology, Huazhong University of Science and Technology, Wuhan 430074, China; d202080693@hust.edu.cn (D.G.); d201980575@hust.edu.cn (J.C.); d201980537@hust.edu.cn (Y.L.); i201921029@hust.edu.cn (C.P.); yangxm@hust.edu.cn (X.Y.); chuqingsun@hust.edu.cn (C.S.); m201971828@hust.edu.cn (J.Y.); 2College of Life Sciences, Henan Normal University, Xinxiang 453007, Henan, China; 15516608057@163.com (X.Z.); jldhwai@163.com (M.J.); 3State Key Laboratory of Infectious Disease Prevention and Control, National Institute for Communicable Disease Control and Prevention, Chinese Center for Disease Control and Prevention, Beijing 102206, China; fanfenxia@icdc.cn

**Keywords:** phage engineering, multidrug resistance, virulence gene, CRISPR–Cas system, tail fiber mutant, nanoparticles, phage immobilization

## Abstract

Along with the excessive use of antibiotics, the emergence and spread of multidrug-resistant bacteria has become a public health problem and a great challenge vis-à-vis the control and treatment of bacterial infections. As the natural predators of bacteria, phages have reattracted researchers’ attentions. Phage therapy is regarded as one of the most promising alternative strategies to fight pathogens in the post-antibiotic era. Recently, genetic and chemical engineering methods have been applied in phage modification. Among them, genetic engineering includes the expression of toxin proteins, modification of host recognition receptors, and interference of bacterial phage-resistant pathways. Chemical engineering, meanwhile, involves crosslinking phage coats with antibiotics, antimicrobial peptides, heavy metal ions, and photothermic matters. Those advances greatly expand the host range of phages and increase their bactericidal efficiency, which sheds light on the application of phage therapy in the control of multidrug-resistant pathogens. This review reports on engineered phages through genetic and chemical approaches. Further, we present the obstacles that this novel antimicrobial has incurred.

## 1. Introduction

In recent years, multidrug-resistant bacterial infections have emerged as one of the most challenging global public health threats, causing severe influences on food safety, environmental ecology, and the social economy [1,2]. Multidrug resistance (MDR) genes derived from gene mutation and gene transfer have also greatly challenged the usage of last-line antibiotic therapy, including carbapenem and polymyxin [3,4]. At present, more than 400 different types and 20,000 potential MDR genes have been predicted from the sequenced bacterial genomic data, which particularly deserves alertness [5]. More seriously, the development of novel antibiotics has become increasingly difficult due to antibiotic resistance. Meanwhile, many pharmaceutical companies have not invested in new antibiotics research and development due to a limited market profit [6]. Therefore, there is a pressing need to search for alternative strategies for MDR bacteria treatment. Phages, natural enemies of bacteria, are considered as a very promising therapy strategy [7]. The first phage research was conducted by Twort, who made an unusual observation about *Staphylococcus* lysis in 1915, which was 13 years before Fleming discovered penicillin, the first antibiotic [8]. Since then, investigators have successively reported many different special viruses in mycoplasmas, spirochetes, actinomycetes, and cyanobacteria [9]. At the onset of phage discovery, they were used to treat bacterial infections, and the initial results were promising [10]. However, the application of phage therapy was limited for its narrow host range and the range of antiviral strategies evolved in bacteria [10,11,12]. Interestingly, phage therapy gradually faded away in the 1940s due to the emergence of antibiotics, yet it has recently regained interest in the wake of the emergence of antibiotic-resistant bacteria. To better cure infections, researchers have developed different phage therapy strategies, including phage cocktails and the combination of phages with other drugs [13,14,15]. More importantly, many researchers have attempted to develop novel engineered phages through genetic and chemical approaches. These artificially armed phages could improve the antibacterial efficacy against MDR bacteria by interfering with their drug-resistant pathways (or in other ways) with little disturbance to the whole microenvironment [16,17]. Here, we summarized the current engineering strategies of phage modification for MDR and discussed the current challenges of phage therapy.

## 2. Phage Genetic Modification

The genetic modification of phages mainly includes the gene mutation, gene replacement, and gene integration of foreign genes using molecular techniques in order to expand the host range or enhance the antibacterial effect of the phages. Gene mutation and replacement usually occur in the genes related to the tail fiber protein to broaden the phage host range. While the integration of foreign genes often integrates some genes into the phage genome nonfunctional region, the products that integrate genes are usually harmful to the host. We summarized the previous articles on the genetic modifications of phages and sorted the results in Table 1 according to the types of phages modified, the modified genes, and the antibacterial mechanisms.

### 2.1. Virulence Gene Overexpression

Phage therapy has been proven to be effective in treating MDR bacterial infections; many clinical case reports support this [13,14,15,29]. However, phage-resistant bacteria are normally generated at a high frequency during treatment, which greatly reduces the efficacy of phage therapy. Some researchers inserted toxin genes into the phage genome (Figure 1), which could be expressed in the host after phage infection, to enhance the bactericidal efficacy.

Hagens et al. [18] inserted *λS105* and Bgl II toxin genes into the genome of phage M13. Phage protein λS105 is a holin protein that could damage the bacterial membrane [30]. Further, the Bgl II restriction enzyme can digest the host chromosome DNA, leading to an irreparable DNA cleavage and cell death (Figure 1A). Both modified phages could kill 99% of *Escherichia coli* MC4100F in less than three hours. Since neither the λS105 protein nor Bgl II restriction enzyme cause cell lysis, there was dramatically less endotoxin detected from the culture medium of the two artificial phages. Westwater et al. [19] also cloned the toxin genes, encoding the proteins Gef and ChpBK into the M13 phage. Toxin Gef and ChpBK could cause bacterial cell membrane damage and mRNA degradation, respectively, thus inducing death of the host bacteria. In vitro studies have shown that an engineered phage expressing Gef or ChpBK reduced the number of live cells of *E. coli* ERAPlacI by 275 times and 370 times, respectively [19].

Antimicrobial peptides (AMPs) are positively charged short peptides with broad-spectrum antibacterial properties [31]. Many AMPs typically antagonize bacteria through multiple simultaneous mechanisms and act on the entire cell membrane, which may be the reason why microorganisms cannot develop a resistance to AMPs as readily as to antibiotics, which usually work on one specific target [32]. Peptide 1018 has been previously reported to have antimicrobial activities against a broad spectrum of bacterial cells and biofilms [33,34]. Lemon et al. [12] integrated the peptide 1018 gene into a bacteriophage T7Select genome. This peptide 1018-producing T7Select not only greatly induced bacteria lysis but also immobilized the extracellular polysaccharide matrix of the bacterial biofilm, along with the release of the phages. Surface-immobilized peptide 1018 continuously killed the nearby cells in the biofilm, causing disintegration of the bacterial biofilm.

### 2.2. MDR System Circumvention

#### 2.2.1. Pathogen-Specific Gene

The extensive and inappropriate use of antibiotics has led to a continuous enrichment of MDR genes in the microbial community [35]. Notably, antibiotics can kill a broad spectrum of microorganisms, regardless of the pathogenic bacteria or commensal bacteria, which causes many antibiotic-dependent diseases, such as *Clostridium difficile* infection (CDI) [36,37]. The strategy of selectively killing MDR bacteria not only eliminates targeted pathogens without disturbing the whole microbial community structure but also hampers the spread of MDR genes and reduces the arsenal of MDR bacterial evolution [16,17].

Bikard et al. [17] cloned a ~2-kb fragment from ΦNM1 against *Staphylococcus aureus (S. aureus)*, including *rinA*, *terS*, *terL*, and other packaging genes, thus ligating this fragment with the CRISPR system (cas9, a tracrRNA sequence) of *Streptococcus pyogenes*, then further inserted guide RNA targeting into the MDR gene to generate MDR-targeting bacteriophages (Figure 2A). The proportion of the subspecies USA300 MRSA of *S. aureus* in the mixed flora significantly decreased from 50% to 0.4% after feeding the engineered phage that targeted the methicillin-resistant gene *mecA*. Furthermore, they made another phage antimicrobial-targeting kanamycin resistance gene, which efficiently reduced the ratio of kanamycin-resistant *S. aureus* from 50% to 10% when used on mouse skin, showing great potential for clinic applications.

Gene *blaNDM-1* and *blaSHV-18* encode an extended spectrum and pan resistance to β-lactam antibiotics, respectively. Citorik et al. [16] constructed RNA-guided nucleases targeting *blaNDM-1* and *blaSHV-18*, respectively, and packaged them into M13 phage DNA named ΦRGN*_ndm-1_* and ΦRGN*_SHV__-18_*. The engineered phages could decrease the population of *E. coli* EMG2, which contains the corresponding resistance genes by two-to-three orders of magnitude, but have no effect on the parent strain itself. Apart from targeting resistance genes, they also designed phages that targeted bacterial virulence genes, such as *eae* of enterohemorrhagic *E. coli* O157: H7 (EHEC). ΦRGN*eae* treatment reduced the EHEC cell number by 20 times in vitro and significantly improved the survival rate of *Galleria mellonella* larvae (*p* < 0.001) infected with EHEC [16]. Similarly, in another study, Park et al. [24] integrated the CRISPR-cas9 system, which targeted the *nuc* gene in the genome of the temperate phage φ sabov to form an engineered phage named φ sabov-cas9-nuc. Bacteriophage φ sabov-cas9-nuc could specifically and efficiently kill *S. aureus* CTH96 strains, both in an in vitro medium culture and an in vivo mouse skin infection model. Taken together, the phage-based antimicrobial agent targeting resistance genes may be a great potential method for curing MDR pathogens.

The Cas9 protein only creates gaps in the chromosome, which can easily be repaired by the host repair system. The Cas3 protein, however, can degrade large fragments from hundreds of bp to 100 kb upstream of the binding sites, which cannot be repaired by the host repair system. Consequently, the Cas3 protein is more efficient than the Cas9 protein in terms of killing bacteria [38]. Yosef et al. [23] cloned the Cas3 gene of the *E. coli* type I CRISPR system into the 19,014-27,480 range of the phage lambda, which did not affect the phage’s adsorption and growth. Afterwards, the guide RNA-targeting resistance genes *NDM-1* (β-lactam resistance) and *ctx-m-15* (carbapenemase resistance) were added (Figure 2B). Compared with the wild-type phages, the engineered λ phage reduced *E. coli* containing *NDM-1* and *ctx-m-15* genes by three orders of magnitude (the number of bacteria changed from 10^7^ to 10^4^ colony-forming units (CFU)) [23].

#### 2.2.2. Biofilm 

Bacteria have two types of models of growth: planktonic cell and sessile aggregates known as biofilms [39]. In the biofilm stage, bacterial cells secrete large amounts of a sticky matrix mainly composed of extracellular polysaccharides, proteins, and DNA, which can enhance bacterial resistance to external environmental pressures, such as reactive oxygen species, antibiotics, zooplankton hunting, and phage invasion [40,41]. Drug-resistant strains resist antibiotics by forming biofilms, which are ubiquitous in communities and hospitals and are frequently detected in water circulation systems, tableware, surgical instruments, injection tubing, and infusion tubing. Researchers have also observed pathogen biofilms in human tissues such as the teeth, ureters, skin, and intestines [42,43,44,45]. Many studies have demonstrated that the long-term applications of antibiotics promote pathogen biofilm development and lead to the accumulation of biofilm-rich mutants in the environment [46,47,48]. Therefore, how to disrupt a biofilm has become a promising direction in the MDR bacteria field.

Itoh et al. [49] cultured several bacteria in a medium containing DspB (50ug/ml) or not and then measured the biofilm formation by crystal violet staining at the indicated times. Their study proved that DspB, a biofilm-degrading enzyme, specifically cleaves β-1, 6-N-acetyl-D-glucosamine, which is important in the stabilization of diverse bacterial biofilms. This caused an almost complete inhibition of the biofilm formation in several different bacteria, such as *E. Coli* MG1655, *E. Coli* TRMG1655, *Staphylococcus epidermidis* 1457, and *Pseudomonas fluorescens* WCS365. Based on their research on DspB, Lu et al. [21] designed an engineered phage, T7_DspB_, by putting a *DspB* gene under the control of a T7φ10 promoter (Figure 1C). DspB was efficiently expressed during the phage infection, causing the degradation of the biofilm. After the treatment with phage T7_DspB_, over 99.997% of the biofilm was removed, and the population of viable bacterial cells in the biofilm decreased by 4.5 orders of magnitude, around 100 times higher than the efficiency of the parent T7 phage. Lu’s work created a novel synthetic biological platform, proving the feasibility of a clear bacterial biofilm using biofilm degradation enzymes and phages.

Quorum-sensing bacteria can produce and secrete some diffusible small molecules named autoinducers, whose concentration can represent the bacterial population density [50]. When the concentration of autoinducers reaches a certain level, they can bind to receptor proteins on a bacterial cell membrane and initiate the expression of a series of genes, regulating many important physiological functions of bacteria, such as biofilm development and virulence gene expression [51]. The phenomenon of bacteria sensing the density of their own and adjusting gene expression is called quorum sensing, which is the language and tool for communication between bacterial cells [52].

Acylhomoserine lactone (AHL) is an autoinducer generally secreted by Gram-negative bacteria and induces the biofilm development of many Gram-negative pathogens [53]. Accordingly, the inhibition of AHL activity could be a promising method for reducing pathogen biofilms. Pei et al. [22] engineered a T7 phage that encoded a lactonase enzyme (AiiA), which degraded the signal molecules AHLs to inhibit quorum sensing, thereby interfering with biofilm formation [22]. Their results showed that Aiia lactase expressed by engineering a T7 phage could effectively degrade AHLs in a variety of bacteria and significantly inhibited the formation of the biofilm in a mixed biofilm composed of *Pseudomonas aeruginosa* and *E. coli*. This quorum-quenching T7 phage may be a promising antibiofilm reagent [22].

#### 2.2.3. SOS System

Antibiotic treatment can damage bacterial chromosomal DNA, producing many single strands of DNA, which can activate the RecA protein. After binding to these single-stranded DNAs, activated RecA degrades LexA, the repressor protein of the SOS response (SOS) repair system, thereby activating the SOS repair system and enabling the bacteria to survive [54]. The SOS repair system can help bacteria develop drug resistances in several ways: (1) increasing the genome mutation rate [55], (2) inducing gene recombinations [56,57], (3) promoting the horizontal transfer of resistance genes [58], and (4) extensively regulating the physiological metabolism of bacteria, such as directly inducing or upregulating the expression of some drug-resistant genes [59]. Interestingly, inactivation of the SOS reaction can enhance the bactericidal effects of antibiotics [60].

Lu et al. [20] inserted the gene encoding the LexA3 protein into the genome of the non-lytic phage M13, which engineered a novel phage called M13mp18. When the M13mp18 phage infected the host, LexA3 was overexpressed, thus inhibiting a SOS reaction (Figure 1B). The results showed that the modified phage could significantly improve the sensitivity of the host bacteria to quinolones, ampicillin, β-lactam, and other antibiotics. M13mp18 increased the bactericidal efficiency of floxacin and ampicillin by 2.7 and 2 orders of magnitude, which were both higher than the antibiotic treatment only. These engineered phages were not only useful for killing wild-type bacteria but also enhanced the killing effect on MDR bacteria. Since the SOS response involves nonessential genes, and the SOS regulatory network is not directly associated with the antibiotic resistance, this strategy can greatly reduce the emergence of engineered phage-resistant strains [20].

### 2.3. Host Range Expansion

#### Tail Fiber Protein

A phage binding to the bacterial surface is the initial step of a successful phage infection, which depends on the crosstalk between the bacterial receptor protein and the phage-binding protein on the tail fiber. Since most phages have unique characteristics (i.e., phage fiber-binding proteins that are different in sizes, shapes, and locations), the host range of most wild-type phages is limited [61]. Mahichi et al. [25] replaced the long fiber gene of T2 with the gene from phage IP008 using homologous recombination (Figure 1D). This engineered phage showed the same host range as IP008 and the strong cleavage activity of T2. Lin et al. [26] genetically modified a T3 phage by replacing a partial tail fiber gene with that from the T7 phage (Figure 1D). Compared with the T3 and T7 wild types, the T3/T7 recombinant phage showed a broader host range and better adsorption efficiency. Marzari et al. [27] designed another chimeric phage expressing a partial receptor-binding domain of phage Ike at the N-terminal of the filamentous coliphage fd G3p. The engineered phage could infect both the natural host bacteria of fd and the host-bearing N-pili of Ike. Compared with wild phages, the infection efficiency of the engineered phage on N-pili *E. coli* increased by 70,000.

However, even the modified phages above have expanded host ranges, and the infected bacteria could still survive through mutagenesis with a high frequency of bacterial receptor proteins. Yehl et al. [28] conducted a high-throughput sequencing analysis of the T3 phage mutant in a coculture system of T3 phage and BL21 (Figure 1D). They found that most mutations occurred at the end of the tail fiber, named the host range determining region (HRDR). Then, they produced a highly diverse phage library of tail fiber proteins via HRDR site mutagenesis. The HRDR mutant library reduced the number of host bacteria by at least five orders of magnitude compared with the wild-type T3. Moreover, the phage library inhibited the in vitro growth of bacteria for ~one week without the emergence of phage resistance, while the control (T3-infected cultures) incurred a significant resistance after ~12 h. Moreover, the similar antimicrobial effect of the phage library was further verified in the in vivo mouse model [28].

## 3. Phage Chemical Modification

The fast growth of the current chemical industry has created numerous novel substances that did not exist before. These substances are entirely new for microorganisms and may function on bacteria with varied mechanisms. In this case, microorganisms have not evolved to generate enough resistance genes to respond to the impact of these new materials [62]. In this section, we will focus on phage chemical modification and summarize our findings in Table 2, wherein several novel chemical materials for modifying phages are presented, including silver nanoparticles (AgNPs), AIEgens, pheophorbide a (PPA), cellulose membrane, and indium tin oxide (ITO).

### 3.1. Phage–Chemical Crosslink

Due to the emergence of multidrug-resistant bacteria, biological agents based on nanoparticles are widely favored [67]. Silver nanoparticles perform excellent bactericide activity simultaneously through various mechanisms, including destroying the bacterial cell wall, inhibiting bacterial enzymes, and binding with DNA, which could interfere with cell division and replication. Dong et al. [63] developed a bacteriophage-based biological and abiotic hybrid reagent to accurately kill *Fusobacterium nucleatum* accumulated in the tumor microenvironment of colorectal cancer in the intestinal tract (Figure 3A). In this study, silver nanoparticles were adsorbed on a coat protein for a filamentous phage M13 through ionic binding. Their results showed that the engineered M13@Ag phage could specifically kill *F. nucleatum* in tumor tissues and, thus, effectively improve the immune microenvironment of the colorectal cancer (CRC) tumor. Meanwhile, the tumor’s growth was inhibited, and the survival time of the tumor-bearing mice was extended.

Photosensitizers with photodynamic inactivation (PDI) activity are the representative achievement of the modern chemical industry development. Photosensitizers can be activated by low-intensity visible light, and the activated photosensitizers can react rapidly with oxygen molecules to produce reactive oxygen species (ROS) [68,69]. ROS can cause a series of reactions in cells, which can damage biological macromolecules such as protein and nucleic acids, destroy bacterial plasma membranes, and induce bacterial or cell death. He et al. [64] linked the photosensitizer AIEgens with photodynamic inactivation (PDI) activity to the surface of phage PAP through an amide bond to form a new type of antibacterial biological conjugate (AIEgens–PAP), which could selectively kill *P. aeruginosa* (Figure 3B). In their study, phages that bound to AIEgens molecules attached themselves precisely to *P. aeruginosa*. Afterwards, under visible light irradiation, AIEgens produced ROS and significantly increased their bactericide efficiency. Besides, the AIEgens–PAP-modified phage also showed excellent performance in the treatment of skin wounds infected with MDR *P. aeruginosa*. Dong et al. [65] conjugated pheophorbide A (PPA), a chlorophyll-based photosensitizer with a JM phage via 1-(3-dimethylaminopropyl)-3-ethylcarbodiimide hydro/N-hydroxysuccinimide (EDC/NHS) crosslinking (Figure 3B). The modified phage could quickly cure skin infected with MDR *Candida albicans* MP65 after light irradiation. The results also demonstrated that, when *C. albicans* MP65 was infected the with PPA-JM phage, the depolarization of the mitochondrial membrane potential decreased, the species and level of the intracellular ROS significantly increased, and mitochondrial dysfunction eventually led to caspase-dependent apoptosis.

### 3.2. Phage Immobilization

Phages are considered ideal materials to target drug-resistant bacteria due to their high specificity. However, phages diffuse easily and are prone to denaturation during the microbicidal process, rendering their antimicrobial efficacy as poor [70]. 

Anany et al. [66] achieved an oriented immobilization of phage cocktails against *E. coli* and *Listeria*, respectively, according to the charge differences between the phage tail fibers and head (Figure 3C). Specifically, the phage head that carries a negative charge could be adsorbed and immobilized onto positively charged cellulose membrane surfaces, especially that which were modified by polyvinylamine, leaving the phage tails free to achieve the highly efficient capture and lysis of the host bacteria. Moreover, the results showed that the immobilized phage cocktails reduced foodborne Listeria on meat surfaces to undetectable levels after treatment for 12 days at 4 ºC. Similar results were obtained for controlling foodborne *E. coli* after treatment for six days at 4 ºC. Compared with the traditional methods for using phages—i.e., phage immersion and spraying—their immobilization approach effectively improved the phage infection efficiency due to the significantly increased phage dose.

In another study, Liana et al. [67] immobilized T4 phages on the surface of indium tin oxide (ITO) through ionic binding to develop the ITO/T4 antimicrobial system (Figure 3C), which could effectively release the T4 phage and show sustained antimicrobial activity [66]. The results showed that the dose of *E. coli* could be reduced by ~10^3^-fold (99.9%) within two hours after being treated by the ITO/T4 system. Then, additional bacteria were added to the first batch; the ITO/T4 systems in the first batch were reduced by ~10^4^-fold in just 30 min, indicating that a single immobilized dose could inhibit the phage for a long time.

## 4. Challenges in Phage Therapy

As the antibiotic resistance crisis becomes more serious, alternative treatments for bacterial infections are urgently needed. Phages are one of the most promising antibacterial agents for clinical use [71]. However, phage therapy is limited by its unstable therapeutic efficacy caused by its narrow host range and bacterial resistance against phages. Interestingly, mounting evidence has demonstrated that engineered phages might be effective for treating bacterial infections. However, the lack of research regarding its effectiveness and safety has naturally limited the clinical or large-scale applications for genetically modified phages. Therefore, the effectiveness and safety tests of engineered phages are urgently needed in further clinical trials.

First of all, phages are strictly host-specific. Accordingly, phage therapy can target certain pathogens without disturbing the gut microbial ecology. However, when the host is infected by various pathogens, single-phage therapy cannot work effectively. Moreover, the sensitive phage needs to be identified according to the isolates that cause the infections, which is a time-consuming process and not suitable for acute infections. A variety of strategies can be used to solve this problem, including the use of a phage cocktail, a modified phage, and the combination of phages and antibiotics.

Secondly, the resistance of bacteria to a phage also results in poor treatment effects of phage therapy. Although some engineering strategies are designed to overcome bacteria resistances to phages and have yielded promising results, bacteria could also evolve themselves to resist those engineered phages gradually, which will thus decrease the efficacy of the engineered phage therapy.

Phages need to reach the infected lesions and adhere to the surface of pathogenic bacteria to exert their functions, so the effective concentrations in the lesions affect their efficacy. In clinical trials, phages cannot reach the lesions or maintain effective concentrations in the lesions due to their limited stability. Besides, most phages could be neutralized by human antibodies [72,73] and cleared by the reticuloendothelial system [74]. In order to ensure that phages can reach the lesions and maintain an effective concentration, it is necessary to determine the time, route, dose, and frequency of administration according to the type of phage and infected bacteria, as well as the different parts of the lesion, so as to determine the best treatment regimen [75]. 

Phages are distinguished from antibiotics and other chemical antimicrobial agents by their self-replicating features, which makes the pharmacokinetics of phages complicated [76]. Therefore, the determination of phage therapy for different pathogens will only become more complex and difficult.

Some improvements have been made for engineered phages to overcome these obstacles [77,78,79], so that engineered phages can reach the lesion sites and maintain effective concentrations in a certain period of time. However, the lack of clinical data is still a serious problem for the widespread application of engineered phages. Therefore, there is an urgent need for future clinical trials to be conducted that evaluate the effectiveness of engineered phages.

The safety of phage therapy cannot be ignored during clinical treatment. Phages can induce immune responses during phage therapy, causing inflammation [80]. The bacterial cell wall fragments, endotoxins, and enterotoxins, which contain impure phage preparations released from the lysis of pathogenic bacteria, may also stimulate the immune system and trigger local acute inflammatory responses [71,81,82,83]. Phages may cause damage to the body, because they carry toxin genes or induce gene mutations [80].

It is also necessary to set standards to evaluate the safety of phage therapy. The Food and Drug Administration (FDA) has set standards based on small-scale experiments that have evaluated the efficacy and safety of phage therapy [84]. Moreover, some standards have been based on the Quality by Design (QbD) and European Union Tissue and Cell Directives (EUTCD), which were established by 32 bacteriophage experts from 12 countries to evaluate the quality and safety of phage therapy products [85]. However, these evaluation systems mainly focus on natural phages. For the development of clinical trials, there is a need to develop a special evaluation standard for the safety evaluation of engineered phages.

In addition to safety and effectiveness, scientists also face challenges when obtaining regulatory approval for phage therapy applications [86]. Currently, there is a lack of standardization and appropriate regulatory framework for phage therapy [87]. Further improvements of the relevant systems and regulations will promote the widespread application of bacteriophages.

## 5. Conclusions

With bacterial resistance becoming a global public health problem, phage therapy is expected to be one of the most promising strategies to treat drug-resistant pathogens. However, phage therapy has not been popularized on a large scale due to its poor therapeutic effects caused by the resistance of bacteria to phages. New genetic engineering and chemical engineering methods are being applied to modify phages, allowing for the more accurate and effective control of drug-resistant bacteria. Although this review mainly focused on applying engineered phages in the antibacterial field, phages can also be used in diagnosis, drug delivery, vaccines, and other fields after a series of modifications. The research on bacteriophages is still in its infancy, although phage engineering has made some progress in enhancing the antibacterial effects of drug-resistant bacteria. Bacteriophages are the most numerous and diverse organisms in nature, which is even more than that of bacteria, but phage modifications have only involved a small percentage of the existing phages, and many phages have not been propagated in a laboratory. Moreover, the genetic manipulation of phages generally depends on the genetic transformation system of the host bacteria, which acts as another obstacle for phage genetic operations. Now, engineered phages, regardless of the engineering methods, are mostly used during in vitro experiments or animal models, but very few clinical trials of phage therapies have been reported. Therefore, a series of safety assessments and large-scale clinical trials need to be conducted before clinical phage applications.

## Figures and Tables

**Figure 1 antibiotics-10-00202-f001:**
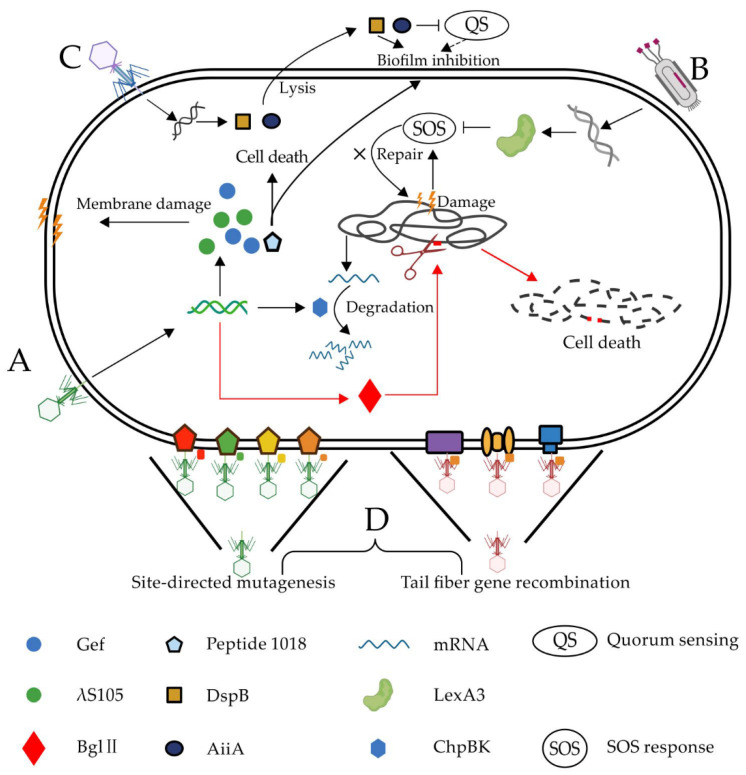
Phage genetic modification strategy against bacteria. Phages are engineered to enhance the bactericidal efficacy by (**A**) expressing toxin proteins, including λ S105, Bgl II, ChpBK, Gef, and peptide 1018; (**B**) suppressing the SOS response (SOS) repair system by overexpressing repressor protein LexA [20]; (**C**) degrading and inhibiting the biofilm using DspB and AiiA, respectively; and (**D**) re-editing the phage tail fiber genes to expand the phage host range [25,26,27,28].

**Figure 2 antibiotics-10-00202-f002:**
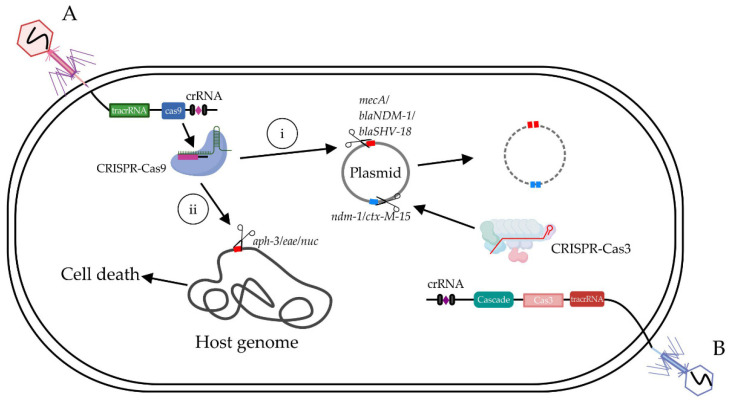
The CRISPR-Cas system including tracrRNA (trans-activating crRNA), Cas9 protein and crRNA (CRISPR RNA) delivered by engineered phages against antibiotic-resistant bacteria. (**A**) Phage-delivered CRISPR-Cas9 could lead to cell death and plasmid loss by targeting genes on the (i) chromosome (*aph-3*/*eae*/*nuc*) and (ii) plasmid (*mecA*/*blaNDM-1*/*blaSHV-18*), respectively [16,17,24]. (**B**) Phage-delivered CRISPR-Cas3 targets specific DNA sequences (*ndm-1*/*ctx-M-15*) in plasmid [23].

**Figure 3 antibiotics-10-00202-f003:**
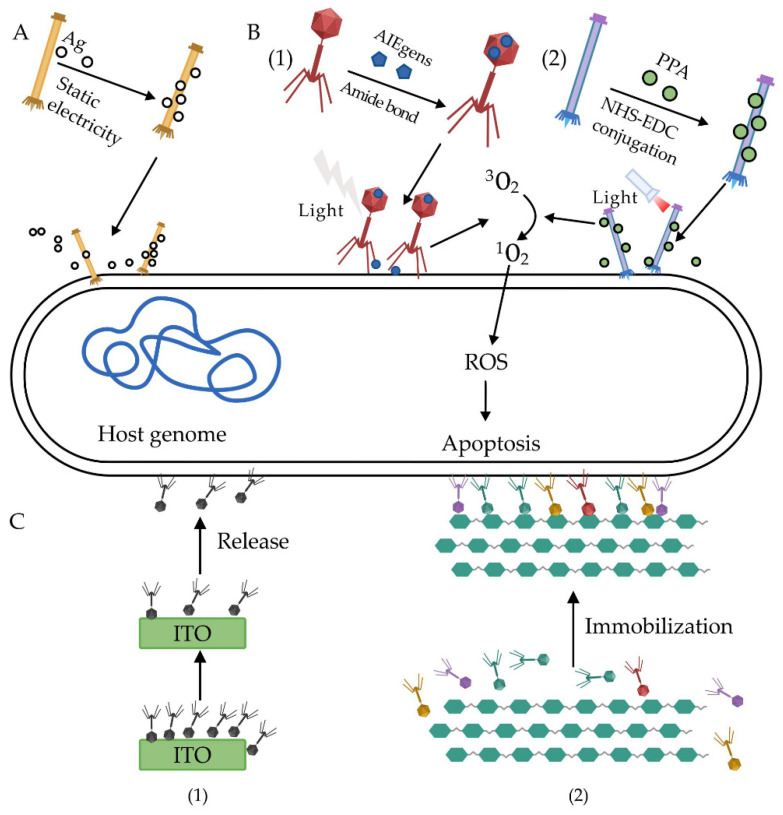
A phage chemical modification strategy against multidrug-resistant (MDR) bacteria. (**A**) Silver nanoparticles delivered by phages directly lead to cell death [63]. (**B**) Photosensitizers AIEgens (1) and pheophorbide a (PPA) (2) loaded on phage can produce cytotoxic reactive oxygen species (ROS) to kill bacteria efficiently after light irradiation [64,65]. (**C**) Phages immobilized on an indium tin oxide (ITO) (1) and cellulose membrane (2) could achieve an efficient inhibition of pathogens due to an increased effective phage concentration [66,67].

**Table 1 antibiotics-10-00202-t001:** Genetic engineering methods applied in phage modification.

Phage	Genetic Modification	Mechanism	Goal/Target	Ref.
M13	λS105; Bgl Ⅱ	Membrane damage; DNA breakage	To reduce endotoxin	[18]
M13	Gef; ChpBK	Membrane damage; mRNA degradation	To increase bactericidal efficiency	[19]
T7Select	peptide 1018	Kill cells; inhibit biofilm	Biofilm	[12]
M13mp18	LexA3	Suppress SOS system	Antibiotic-resistant bacteria	[20]
Wild-type T7	DspB	Hydrolysis β-1,6-*N*-acetyl-d-glucosamine	Biofilm	[21]
T7Select415-1	AiiA	Inhibit quorum sensing	Biofilm	[22]
M13 phagemid	CRISPR-cas9	Target resistance genes	Antibiotic-resistant bacteria	[17]
M13 phagemid	CRISPR-cas9	Target resistance genes and virulent genes	Antibiotic-resistant bacteria	[16]
λ phage	CRISPR-cas3	Target resistance genes	Antibiotic-resistant bacteria	[23]
φ SaBov	CRISPR-cas9	Target the nuc gene	Antibiotic-resistant bacteria	[24]
T2, T3, Fd	Tail fiber genes	Expand the host range	Antiphage bacteria	[25,26,27,28]

**Table 2 antibiotics-10-00202-t002:** Chemical engineering methods applied for phage modification. EDC/NHS: 1-(3-dimethylaminopropyl)-3-ethylcarbodiimide hydro/N-hydroxysuccinimide.

Phage	Chemical Modification	Binding Force	Ref
M13	Silver nanoparticles (AgNPs)	Ionic binding	[63]
PAP	AIEgens	Amide bond	[64]
JM phage	Pheophorbide a (PPA)	EDC/NHS Crosslinking	[65]
Bacteriophage T4 (ATCC 11303-B4)	Indium tin oxide (ITO)	Ionic binding	[66]
Phage cocktail	Cellulose membrane	Ionic binding	[67]

## Data Availability

No new data were created or analyzed in this study. Data sharing is not applicable to this article.

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
