# Peer review of "Genetic and Chemical Engineering of Phages for Controlling Multidrug-Resistant Bacteria"

_antibiotics, 2021, doi:10.3390/antibiotics10020202_

Round 1

Reviewer 1 Report

Overall, I found this to be a reasonable manuscript.  It will require major editing to make it more suitable for publication.

While I am favorable to the manuscript, I think the authors need to create a future directions section and expand on the hurdles facing phage therapy.  Phages are not a panacea to MDR-bacteria. If they were, they would be used more widely.  There are many reasons why phages have not become first line therapies. For example, there needs to be some discussion regarding the need to define optimal (1) route of administration, (2) dose and frequency, and (3) duration of therapy. These are some discussion regarding the issues with delivering phage therapy (they have limited stability and denature easily).  Phages are not active against all strains and many bacteria development resistance rapidly.  There needs to be a discussion on how there is a need to identify the optimal ways to use phages and antibiotics in conjugation.  Currently, phages are largely used after patients fail antibiotics and the benefits of phages are earlier in the course of therapy.  early use is what is modelled in in vitro and animal studies but this is not how they are used in practice.  Nearly all the data presented in this paper involved in vitro or animal experiments and I encourage the authors to include more human data/experiences with some of the novel phage therapies.  There also needs to be a discussion of the regulatory hurdles with phage therapy, and there needs to be a review of some of the failed phage clinical trials and lessons learned.   

Below are some minor comments.  As stated above, the manuscript needs substantial editing as it is awkwardly written in most places. 

  1. Line 31: reads awkwardly and it not entirely accurate. Much of resistance is chromosomally mediated.
  2. Lines 45-54: needs to be re-written as it is unclear and not entirely accurate.
  3. Line 67: a better supportive reference is needed.
  4. Figure 1: needs better detailing in section 2.1
  5. Line 77: needs a better transition as the transition from the previous paragraph is abrupt.
  6. Lines 77-86: is very choppy
  7. Line 88: I am not sure how AMPs fit here. Perhaps it should be the start of a new paragraph.  The statement is not entirely accurate as bacteria readily develop resistance to naturally occurring AMPs.
  8. Section 2.2.1: does not read well and would benefit from substantial editing.
  9. Line 104: the wording does not make sense and needs to be re-written.
  10. Need a reference for line 109
  11. If available, line 138 would benefit from inclusion of any available human data.
  12. Line 162: more detailing of the experiments cited are needed. It is unclear how the experiment was performed (i.e., conditions)
  13. Line 171: reads awkwardly.

Author Response

While I am favorable to the manuscript, I think the authors need to create a future directions section and expand on the hurdles facing phage therapy.  Phages are not a panacea to MDR-bacteria. If they were, they would be used more widely.  There are many reasons why phages have not become first line therapies. For example, there needs to be some discussion regarding the need to define optimal (1) route of administration, (2) dose and frequency, and (3) duration of therapy. These are some discussion regarding the issues with delivering phage therapy (they have limited stability and denature easily).  Phages are not active against all strains and many bacteria development resistance rapidly.  There needs to be a discussion on how there is a need to identify the optimal ways to use phages and antibiotics in conjugation.  Currently, phages are largely used after patients fail antibiotics and the benefits of phages are earlier in the course of therapy.  early use is what is modelled in in vitro and animal studies but this is not how they are used in practice.  Nearly all the data presented in this paper involved in vitro or animal experiments and I encourage the authors to include more human data/experiences with some of the novel phage therapies.  There also needs to be a discussion of the regulatory hurdles with phage therapy, and there needs to be a review of some of the failed phage clinical trials and lessons learned. 

Response: Thank you for your time and effort considering our manuscript. We do our best to search the relevant literature, but we feel sorry that there are few articles about phage chemical modification. So we failed to develop this section on chemical modifications. In the current version, we have added a section about the limitations in the clinical application of phage therapy. And we also give some suggestions on how to address these barriers. For several parts that is weakly written, we have done our best to make a revision. Finally, we have also reinserted the figure. Thanks again for your valuable comments.

Line 31: reads awkwardly and it not entirely accurate. Much of resistance is chromosomally mediated.

Response: We appreciate referee’s comment. We have re-write the sentences. Please see line 36-38.

Lines 45-54: needs to be re-written as it is unclear and not entirely accurate.

Response: We agree. We have re-write this part and corrected some errors. Please see line 49-58.

Line 67: a better supportive reference is needed.

Response: We agree. We have cited the appropriate reference. Please see line 74.

Line 77: needs a better transition as the transition from the previous paragraph is abrupt.

Response: We agree. We have revised this section. Please see line 84.

Lines 77-86: is very choppy

Response: We agree. We have revised this section. Please see line 84-95.

Line 88: I am not sure how AMPs fit here. Perhaps it should be the start of a new paragraph.  The statement is not entirely accurate as bacteria readily develop resistance to naturally occurring AMPs.

Response: 1) Thanks. We have revised.

2) We cited the reference 29 and we think the appropriate statement about AMPs may be “Many AMPs typically antagonize bacteria through multiple simultaneous mechanisms and act on the entire cell membrane, which may be the reason that microorganisms can not develop resistance to AMPs as readily as to antibiotics which usually work on one specific target”. Please see line 97-100.

Section 2.2.1: does not read well and would benefit from substantial editing.

Response: We agree. We have checked this part carefully and corrected some possible inappropriate descriptions in the revised manuscript.

Line 104: the wording does not make sense and needs to be re-written.

Response: We agree. We have modified the sentence. Please see line 111-114.

Line 109: Need a reference for line 109.

Response: We have added the appropriate reference. Please see line 116.

Line 38: If available, line 138 would benefit from inclusion of any available human data.

Response: We agree. But unfortunately, we found it was difficult to find useful information. We are sorry for that.

Line 162: more detailing of the experiments cited are needed. It is unclear how the experiment was performed (i.e., conditions)

Response: We have added relevant details to complement this experiment. Please see line 170-176.

Line 171: reads awkwardly.

Response: We agree. We have rewritten the sentence. Please see line 184-185.

Reviewer 2 Report

The aim of the minireview entitled “Genetic and Chemical Engineering on Phage for Multi-drug Resistant Pathogen Treatment” by Guo et al.  is to summarize the most recent progresses in this very interesting and promising field of research. The review is well organized, but information presented is rather limited, i.e., the review is very short, English language usage needs very substantial improvement, figures lack any information on the sources used to generate them (references), citations are done incorrectly in many places, references are also not correctly placed, and in a few instances, references do not correspond to the topic discussed. Overall, the work, though promising, is very superficial and needs serious improving.

Corrections and additions are needed throughout the MS. Below only a few are mentioned.

Title is not correct. Why “on phage”? It should be “of Phages”. Why “Multi-drug Resistant Pathogen Treatment”. The approach aims at controlling MDR bacteria with the aid of phages, and to provide an alternative treatment to antibiotics. “Multi-drug Resistant Pathogen Treatment” fails to express these facts.

Lines 43

Change investigator with investigators

Lines 45-46

“Since phage was discovered, it has been used to treat bacterial infections and its initial results were

promising”

This sentence should use the plural. “Since phages were discovered, they have been used to treat bacterial infections and its the initial results were promising”.

Line 46

Remove capital T from The CRISPR-based …”

Line 48

“Interestingly, in the old time, the discovery of antibiotics eliminated …”

What do authors understand by “in the old time …”. In ancient times? This is unproper scientific writing.

Lines 50-52

Rewrite the sentence “For better cure infections and reduce the emergence of phage resistant bacteria … “

At minimum it should read something like “For better cure of infections and to reduce the emergence of phage resistant bacteria  …”

Line 56-58

Make correction to the sentence containing “by interfering their drug-resistance pathways or other ways …”. It could be something like “by interfering with their drug-resistance pathways or in other ways …”

Line 62-64.

This section has to be developed and data summarized in Table 1 should be explained.

Line 70

Figure 1 in the MS is distorted. The reviewer knows that this might be cause by how the PDF file of the MS was generated. However, in the current form the content of this figure cannot be reviewed. Also, the legend lacks the references that contain the information used by authors to generate figure 1. The assumption is that this is a novel figure.

Line 105

clostridium difficile should be Clostridium difficile.

Line 106

Specifically should be specifically

Lines 119 and 270

Figures 2 and 3. These figures have the same issues as Figure 1.

Lines 161-163.

The sentence is unclear.

DspB, a biofilm degrading enzyme, has been proved to degrade several different bacterial biofilms by hydrolyzing β-1, 6-N-acetyl-D-glucosamine, like E.Coli K-12 MG1655, E.Coli TRMG1655, Staphylococcus epidermidis 1457 and Pseudomonas fluorescens, WCS365.

Lines 163-164.

Citation is not done correctly and reference missing.  ”Timothy K. Lu et al. designed an engineering phage T7 DspB via putting DspB gene under control of T7φ10 promoter.”. It should be: “Lu et al. (reference#) designed ….”

Lines 168-170.

Citation is not done correctly, and reference is wrong. Reference 40 is about the work of Itoh et al. Rewrite the sentence and correct the reference. “Timothy K. Lu’s work created a novel synthetic biological platform, proved the feasibility of clear bacterial biofilm using biofilm-degradation enzyme and phage [40]. “

Line 251.

Phage Chemical Modification should be Phage chemical modification. This section should be developed, and the limited information presented in table 2 should be discussed.

Line 254.  Phage-chemical Crosslink should be Phage-chemical crosslink.

Line 263

Add the references after Zang et al. (reference#)

Line 300

Incorrect citation: H.Anany et al. Remove H. Add reference after Anany et al. (reference#)

Line 310

Incorrect citation: Ayu E. Liana et al. Add reference#

Author Response

The aim of the minireview entitled “Genetic and Chemical Engineering on Phage for Multi-drug Resistant Pathogen Treatment” by Guo et al.  is to summarize the most recent progresses in this very interesting and promising field of research. The review is well organized, but information presented is rather limited, i.e., the review is very short, English language usage needs very substantial improvement, figures lack any information on the sources used to generate them (references), citations are done incorrectly in many places, references are also not correctly placed, and in a few instances, references do not correspond to the topic discussed. Overall, the work, though promising, is very superficial and needs serious improving.

Response: Thank you for your time and effort considering our manuscript. We agree the information present is limited, and we have added a section about the barriers in the clinical application of phage therapy. And we have given some suggestions on how to address these barriers. We hope that the additions will enrich this review and improve the quality of the manuscript. In addition, we have checked our manuscript carefully and make some improvements in the English language usage and references which were cited inappropriately. We have also added related information about the sources of the figures. Thanks again for your valuable comments.

Corrections and additions are needed throughout the MS. Below only a few are mentioned.

Title is not correct. Why “on phage”? It should be “of Phages”. Why “Multi-drug Resistant Pathogen Treatment”. The approach aims at controlling MDR bacteria with the aid of phages, and to provide an alternative treatment to antibiotics. “Multi-drug Resistant Pathogen Treatment” fails to express these facts.

Response: We appreciate your comment. We have revised the title.

Lines 43

Change investigator with investigators.

Response: Thanks. We have revised as required.

Lines 45-46

“Since phage was discovered, it has been used to treat bacterial infections and its initial results were promising” This sentence should use the plural. “Since phages were discovered, they have been used to treat bacterial infections and its the initial results were promising”. 

Response: Thanks. We have rewritten the sentence. Please see line 49-51.

Line 46

Remove capital T from The CRISPR-based …”

Response: Thanks. We have revised.

 Line 48

“Interestingly, in the old time, the discovery of antibiotics eliminated …” What do authors understand by “in the old time …”. In ancient times? This is unproper scientific writing.

Response: Thanks for referee’s comments. We have rewritten the sentence. Please see line 52-54.

Lines 50-52

Rewrite the sentence “For better cure infections and reduce the emergence of phage resistant bacteria … “ At minimum it should read something like “For better cure of infections and to reduce the emergence of phage resistant bacteria  …” 

Response: We appreciate referee’s comment. We have rewrite this sentence. Please see line 54-56.

Line 56-58

Make correction to the sentence containing “by interfering their drug-resistance pathways or other ways …”. It could be something like “by interfering with their drug-resistance pathways or in other ways …”

Response: We appreciate referee’s comment. We have revised as required.

 Line 62-64.

This section has to be developed and data summarized in Table 1 should be explained ?

Response: We agree. We have developed this part. Please see line 64-71.

Line 70

Figure 1 in the MS is distorted. The reviewer knows that this might be cause by how the PDF file of the MS was generated. However, in the current form the content of this figure cannot be reviewed. Also, the legend lacks the references that contain the information used by authors to generate figure 1. The assumption is that this is a novel figure.

Response: We have re-inserted the figure and added the information about the sources of the figures.

 Line 105

clostridium difficile should be Clostridium difficile.

Response: Thanks. We have revised as required.

Line 106

Specifically should be specifically

Response: Thanks. We have revised as required.

Lines 119 and 270

Figures 2 and 3. These figures have the same issues as Figure 1.

Response: Thanks. We have revised.

Lines 161-163.

The sentence is unclear.

DspB, a biofilm degrading enzyme, has been proved to degrade several different bacterial biofilms by hydrolyzing β-1, 6-N-acetyl-D-glucosamine, like E.Coli K-12 MG1655, E.Coli TRMG1655, Staphylococcus epidermidis 1457 and Pseudomonas fluorescens, WCS365. 

Response: We agree. We have added a couple of sentences explaining in more detail the sentence. Please see line 170-176.

Lines 163-164.

Citation is not done correctly and reference missing.  ”Timothy K. Lu et al. designed an engineering phage T7 DspB via putting DspB gene under control of T7φ10 promoter.”. It should be: “Lu et al. (reference#) designed ….”

Response: We agree. We have corrected the reference. Please see line 177.

We have Lines 168-170.

Citation is not done correctly, and reference is wrong. Reference 40 is about the work of Itoh et al. Rewrite the sentence and correct the reference. “Timothy K. Lu’s work created a novel synthetic biological platform, proved the feasibility of clear bacterial biofilm using biofilm-degradation enzyme and phage [40]. “ 

Response: We appreciate referee’s comment. We have modified the reference.

Line 251.

Phage Chemical Modification should be Phage chemical modification. This section should be developed, and the limited information presented in table 2 should be discussed.

Response: Thanks. 1)We have modified Phage Chemical Modification to Phage chemical modification. 2) We agree that Phage Chemical Modification lacks the contents. We’ve tried our best to search the relevant literature, but we feel sorry that there are few articles about phage chemical modification. So we failed to develop this section on chemical modifications. 3) We have briefly described the contents of the table 2. If the referee still feels that the contents of Table 2 have not been adequately discussed, we will make subsequent revision.

Line 254. 

Phage-chemical Crosslink should be Phage-chemical crosslink.

Response: Thanks. We have revised as required.

Line 263

Add the references after Zang et al. (reference#) 

Response: Thanks. We have revised as required.

Line 300

Incorrect citation: H.Anany et al. Remove H. Add reference after Anany et al. (reference#) 

Response: Thanks. We have revised as required.

Line 310

Incorrect citation: Ayu E. Liana et al. Add reference#

Response: Thanks for referee’s comments. We have revised.

Reviewer 3 Report

Comments to Author

The authors described current strategies of phage engineering for the therapeutic application. The manuscript addressed various approaches and the topic is quite interesting for the readers of Antibiotics, however, it is quite restricted to particular area; Genetic engineering focused on CRISPR and chemical engineering lacks the contents. And it is better to include the limitations of each approaches. In addition, manuscript in several part is weakly written and not well documented. Definitely, the quality of figures should be upgraded.

Line 45: please clarify the sentence.

Line 47-48: Is this a single mechanism of phage resistance that can represent the referred study?

Line 53-55: The reference 14 is a very good paper, unfortunately, it do not support this statement. The research only support for the phage resistant mechanism.

Line 55-58: The authors stated the phage resistance until line 55. However, the conclusion is a bit on the wrong way. What is the context of the manuscript; How can the modifications improve the (1) phages’ antibacterial effect? (2) phage resistance? If authors wanted to say in the first way, the previous sentences should be written in different way. There is no correlation between the sentences before and after line55. 

Line 66: I barely see the “many clinical reports” from the reference. In addition, that research is published at “Experimental Therapeutics” part. I recommend referring the study addressing more broad area, not specific.

Line 69: Please increase the resolution of the figure. I can not properly get the information in the figure.

Line 77: I can not get why this sentence was located in this section.

Line 79-88: It is better to referring in the numerical order.

Line 98 (Table1): Authors should provide the correct information for the readers. The mechanism of peptide 1018 is not specifically targeting the ppGpp as stated in the reference 26. Study [33],[53~56] examined other phage in their study. Authors should include it. And study [59] used CRISPR-cas mechanism for the modification and used methyltransferase of RM system from Streptococcus for the selection. / Please use the proper reference style for 45 at the reference section. / And for the quality of the paper, I recommend to supplement the “goal/target” of the study, i.e. biofilm, antibiotic resistant bacteria, phage resistant bacteria, in the table.

Line 102: The reference [27] is not appropriate for the sentence. Topic of the study is detection of the antibiotic resistance, not the emerging of MDR genes caused by extensive use of antibiotics.

Line 103-106: Please refer proper references. Very broad statement with very specific and not related references.

Line 119-123: As previous comment, please increase the resolution.

Line 124: Please unify the use of “β – lactam” through the manuscript for the quality of the manuscript. Somewhere it is written with no space.

Line 126: It looks like there is space in “ΦRGN SHV-18".

Line 140: There should be space in “100kb”

Line 147: Please include the unit after the bacterial count.

Line 150: Do authors really think the reference [34, Antimicrobial resistance of Pseudomonas aeruginosa biofilms.] is supporting the bacterial two life style; planktonic and sessile?

Line 156-157: In my opinion, they did not observe pathogen biofilm in human tissues in that referred research [37, Dispersing biofilms with engineered enzymatic bacteriophage.]

Line157-158: Please use the supporting reference. This is very specific and I can't find "many” studies.

Line 164-170: “T7 DspB” not “T7DspB”? And provide the proper reference number.

Line 171-175: Please provide the reference(s).

Line 180-181: Where is the report the authors mentioned?

Line 184-186: It is second time to use “Pseudomonas”. Better to shorten. And space is needed for “P.aeruginosa, and E.coli”.

Line 211-212: This sentence should be more clear. There are plenty of phages having broad host range, even crossing the genus. And the main theme of the referred research is the "tunable host range” with the engineering. Again, please provide the correct information.

Line 235 (Talbe2): gold nanoparticles also examined in [63]. Reference [69] is not for ITO modification and where is [71]? I thought the authors confused the reference order. Again, please refer in numerical order.

Line 237: Check the numbering of the heading. And unify the term "R-M system” through the manuscript for the quality.

Line 243: Please unify the term "CRISPR Cas system” through the manuscript.

Line 238-250: Reference [57] is a research about database, not for the mechanism of RM system as mentioned in the sentence. And the contents about RM system sounds plausible. However, the referred paper [59] used the "methyltransferase of other bacteria's RM system as it does not interact with the examined bacteria. SEE THE MANUSCRIPT: [We elected to introduce a methyltransferase gene of the type II restriction/modification (R/M) system LlaDCHI (59) from L. lactis since this system is functional in S. thermophilus DGCC7710 and does not interfere with the DGCC7710 CRISPR-Cas systems.] Manuscript should be written based on the findings, not the speculations.

The statement in line 247-248 “The engineered phage smq-1107 almost kill 100% host bacteria, while the phage plaque-forming rate of its wild-type phage is only 10-6.” needs to be revised with the firm understanding of “EOP”. “EOP 1” does not mean the phages kill 100% of host bacteria.

Line 270: Quality of figure should be increased.

Line276-295: “P.aeruginosa” space. Better to use C. albicans as you used abbreviation in other microorganism for the quality of manuscript.

Line 300-309: “H.Anany et al.”, “E.coli”, and “4℃” need space. Please space before the units through the manuscript.    

Reference: REFER PROPER references. And formatting should be done; scientific name, abbreviation, italic style.

Author Response

The authors described current strategies of phage engineering for the therapeutic application. The manuscript addressed various approaches and the topic is quite interesting for the readers of Antibiotics, however, it is quite restricted to particular area; Genetic engineering focused on CRISPR and chemical engineering lacks the contents. And it is better to include the limitations of each approaches. In addition, manuscript in several part is weakly written and not well documented. Definitely, the quality of figures should be upgraded.

Response: Thank you for your time and effort considering our manuscript. We agree with you and think that it is a major drawback our genetic engineering focused on CRISPR and chemical engineering lacks the contents. We’ve tried our best to search the relevant literature, but we feel sorry that there are few articles about phage chemical modification. So we failed to develop this section on chemical modifications. In the current version, we have added a section about the limitations in the clinical application of phage therapy. And we also give some suggestions on how to address these barriers. For several parts that is weakly written, we have done our best to make a revision. Finally, we have also reinserted the figure. Thanks again for your valuable comments!

Line 45: please clarify the sentence.

Response: We mean that phages were used early on as antimicrobial agent. And their initial results of therapy were promising. We have rewritten the sentence, please see line 49-51.

Line 47-48: Is this a single mechanism of phage resistance that can represent the referred study?

Response: We appreciate referee’s comment. We have rewritten this sentence,please see line 51-52.

Line 53-55: The reference 14 is a very good paper, unfortunately, it do not support this statement. The research only support for the phage resistant mechanism.

Response: We agree. For the sake of contextual logic and coherence, we have removed this sentence.

Line 55-58: The authors stated the phage resistance until line 55. However, the conclusion is a bit on the wrong way. What is the context of the manuscript; How can the modifications improve the (1) phages’ antibacterial effect? (2) phage resistance? If authors wanted to say in the first way, the previous sentences should be written in different way. There is no correlation between the sentences before and after line55. 

Response: We appreciate referee’s comment. We have revised this part. Please see line 54-60.

Line 66: I barely see the “many clinical reports” from the reference. In addition, that research is published at “Experimental Therapeutics” part. I recommend referring the study addressing more broad area, not specific.

Response: We agree. In the new version, we have cited several appropriate and representative references.

Line 69: Please increase the resolution of the figure. I can not properly get the information in the figure.

Response: We feel sorry for that. We have reinserted the figure.

Line 77: I can not get why this sentence was located in this section.

Response: Thanks. We have revised this part, please see line 84-87.

Line 79-88: It is better to referring in the numerical order.

Response: Thanks for referee’s comments. We have checked and made a revision in the full text.

Line 98 (Table1): Authors should provide the correct information for the readers. The mechanism of peptide 1018 is not specifically targeting the ppGpp as stated in the reference 26. Study [33],[53~56] examined other phage in their study. Authors should include it. And study [59] used CRISPR-cas mechanism for the modification and used methyltransferase of RM system from Streptococcus for the selection. / Please use the proper reference style for 45 at the reference section. / And for the quality of the paper, I recommend to supplement the “goal/target” of the study,i.e. biofilm, antibiotic resistant bacteria, phage resistant bacteria, in the table.

Response:1)We appreciate referee’s comments. We have made a correction to the mechanism of peptide 1018.

2)Study [33],[53~56] do involve other phages. However, we only show in the table the phages that were successfully engineered to improve bactericidal efficiency, i.e. the end products of the modification.

3)We read the reference in detail and found that we did make a mistake there. Their study used methyltransferase of RM system just for the selection. So the reference is not relevant to our topic. In order to improve the quality of documents, we removed this part.

4) We appreciate your comments. We have revised.

Line 102: The reference [27] is not appropriate for the sentence. Topic of the study is detection of the antibiotic resistance, not the emerging of MDR genes caused by extensive use of antibiotics.

Response: We agree. We have cited more appropriate reference to support the sentence.

Line 103-106: Please refer proper references. Very broad statement with very specific and not related references.

Response: We agree. We have revised the sentence and refer proper references. Please see line 111-114.

Line 119-123: As previous comment, please increase the resolution.

Response: We have reinserted the figure.

Line 124: Please unify the use of “β – lactam” through the manuscript for the quality of the manuscript. Somewhere it is written with no space.

Response: We have revised as required.

Line 126: It looks like there is space in “ΦRGN SHV-18".

Response: We agree. We have revised.

Line 140: There should be space in “100kb”

Response: We agree. We have revised as required.

Line 147: Please include the unit after the bacterial count.

Response: Thanks for your comments. We have revised.

Line 150: Do authors really think the reference [34, Antimicrobial resistance of Pseudomonas aeruginosa biofilms.] is supporting the bacterial two life style; planktonic and sessile?

Response: We appreciate your comments. We have referred the appropriate reference.

Line 156-157: In my opinion, they did not observe pathogen biofilm in human tissues in that referred research [37, Dispersing biofilms with engineered enzymatic bacteriophage.]

Response: We appreciate your comments. We have referred the appropriate references.

Line157-158: Please use the supporting reference. This is very specific and I can't find "many” studies.

Response: We agree. We have referred the appropriate references

Line 164-170: “T7 DspB” not “T7DspB”? And provide the proper reference number.

Response: T7 DspB and T7DspB and are identical. We have corrected the reference number.

Line 171-175: Please provide the reference(s).

Response: We have provided the references.

Line 180-181: Where is the report the authors mentioned?

Response: We have revised the sentence and added reference. Please see line 194-196.

Line 184-186: It is second time to use “Pseudomonas”. Better to shorten. And space is needed for “P.aeruginosa, and E.coli”.

Response: Thanks. We have corrected them.

Line 211-212: This sentence should be more clear. There are plenty of phages having broad host range, even crossing the genus. And the main theme of the referred research is the "tunable host range” with the engineering. Again, please provide the correct information.

Response: We cited the reference 64 and we think the appropriate statement may be “the host range of most wild-type phages is limited”, please see line 227-229.

Line 235 (Talbe2): gold nanoparticles also examined in [63]. Reference [69] is not for ITO modification and where is [71]? I thought the authors confused the reference order. Again, please refer in numerical order.

Response:

1) Thanks for your comments. We think gold nanoparticles were used in the study63 only as a control to highlight the better bactericidal effect of silver nanoparticle. Moreover, gold nanoparticles were not used to modify the phage. So we didn't discuss it.

2) Thanks a lot. We did confuse the reference order. We have referenced in numerical order.

Line 237: Check the numbering of the heading. And unify the term "R-M system” through the manuscript for the quality.

Response: We have removed this part.

Line 243: Please unify the term "CRISPR Cas system” through the manuscript.

Response: We have unified the term "CRISPR-Cas system” through the manuscript.

Line 238-250: Reference [57] is a research about database, not for the mechanism of RM system as mentioned in the sentence. And the contents about RM system sounds plausible. However, the referred paper [59] used the "methyltransferase of other bacteria's RM system as it does not interact with the examined bacteria. SEE THE MANUSCRIPT: [We elected to introduce a methyltransferase gene of the type II restriction/modification (R/M) system LlaDCHI (59) from L. lactis since this system is functional in S. thermophilus DGCC7710 and does not interfere with the DGCC7710 CRISPR-Cas systems.] Manuscript should be written based on the findings, not the speculations.

Response: We read the reference in detail and found that we did make a mistake there. Their study used methyltransferase of RM system just for the selection. So the reference is not relevant to our topic. In order to improve the quality of documents, we removed this part.

The statement in line 247-248 “The engineered phage smq-1107 almost kill 100% host bacteria, while the phage plaque-forming rate of its wild-type phage is only 10-6.” needs to be revised with the firm understanding of “EOP”. “EOP 1” does not mean the phages kill 100% of host bacteria.

Response: We appreciate your comments. We have removed this part.

Line 270: Quality of figure should be increased.

Response: We agree. We have inserted new figure.

Line276-295: “P.aeruginosa” space. Better to use C. albicans as you used abbreviation in other microorganism for the quality of manuscript.

Response: Thanks. We have revised.

Line 300-309: “H.Anany et al.”, “E.coli”, and “4℃” need space. Please space before the units through the manuscript.    

Response: Thanks. We have revised.

Reference: REFER PROPER references. And formatting should be done; scientific name, abbreviation, italic style.

Response: We appreciate your comments. We have checked references throughout the MS and corrected those that were problematic.

Round 2

Reviewer 1 Report

NA

Author Response

Reviewer #1

Comments: NA

Dear referee, we have the manuscript corrected by a native English speaker this time. And we think it may be a better version now. We will continue to work hard afterwards and try to complete a better paper. Thanks again for the valuable comments.

Reviewer 2 Report

The review entitled “Genetic and Chemical Engineering of Phage for Multi-drug Resistant Bacteria controlling” by Guo et al.  has been improved but not sufficiently to be published as such. The authors added more information as the review was very short and provided several citations to the legend of figures. However, English language still needs to be improved in several places and citations are not done properly in many instances in spite of the fact that this issue was mentioned in the previous comments.

Specific comments (only some issues are mentioned, the authors should try to further improve English language throughout the review)

Title

The title is still not written in proper English: Genetic and Chemical Engineering of Phage for Multi-drug Resistant Bacteria controlling

A possible option would be:

Genetic and Chemical Engineering of Phage for controlling Multi-drug Resistant Bacteria

Citation of references in many places is still incorrect. It is very surprising that so many authors could not figure out that citation like “author et al (year)” should mention only family name, then et al., and then the year.

For example, line 82

“S. Hagens et al. [25]” should be just “Hagens et al. [25]”

Line 88

“Caroline Westwater et al. [27]” should be “Westwater et al. [27]”

The same mistakes, quite annoying as this issue was indicated in the previous review, are in lines 100, 115, 129, 137, 148, 167, 173, 179, 210, 227 (family name should start with a capital letter), 230, 233, 240 (family name should start with a capital letter), 285, 293, 316. Maybe there are other that I missed …

Line 77. Phage is engineered for enhancing the bactericidal efficacy by

Suggest using plural: Phages are engineered for ….

Line 279

Suggest replacing “sustained” with something like “increased”

Line 287

Amide bond should be amide bond

Line 297

C. Albicans should be C. albicans

Line 349 due to their unstable.

Suggest replacing with something like: “due to their limited stability”

Line 362 Therefore, we urged that more clinical trials should be done in the future ….

Suggest replacing with something like:  Therefore, there is an urgent need for more clinical trials to be conducted in the future …

Line 367 which contained

 Add are. Which are contained

Line 270

Ionic binding should be ionic binding (no capital letter for ionic)

Line 377 It needs,

Suggest replacing with something like: There is a need

Author Response

Reviewer #2

Comments: The review entitled “Genetic and Chemical Engineering of Phage for Multi-drug Resistant Bacteria controlling” by Guo et al.  has been improved but not sufficiently to be published as such. The authors added more information as the review was very short and provided several citations to the legend of figures. However, English language still needs to be improved in several places and citations are not done properly in many instances in spite of the fact that this issue was mentioned in the previous comments.

Thanks for your valuable comments. We feel deeply sorry for that we were not aware of the incorrect citation in the last revision because we lacked the related experience. We have revised them throughout the manuscript this time. In addition, we have our manuscript corrected by a native English speaker. We think it maybe a better version now. Thanks again for the valuable comments.

Specific comments (only some issues are mentioned, the authors should try to further improve English language throughout the review)

1)Title

The title is still not written in proper English: Genetic and Chemical Engineering of Phage for Multi-drug Resistant Bacteria controlling

A possible option would be:

Genetic and Chemical Engineering of Phage for controlling Multi-drug Resistant Bacteria

Response: We appreciate referee’s comment. We have revised the title.

2) Citation of references in many places is still incorrect. It is very surprising that so many authors could not figure out that citation like “author et al (year)” should mention only family name, then et al., and then the year.

For example, line 82

“S. Hagens et al. [25]” should be just “Hagens et al. [25]”

Line 88

“Caroline Westwater et al. [27]” should be “Westwater et al. [27]”

The same mistakes, quite annoying as this issue was indicated in the previous review, are in lines 100, 115, 129, 137, 148, 167, 173, 179, 210, 227 (family name should start with a capital letter), 230, 233, 240 (family name should start with a capital letter), 285, 293, 316. Maybe there are other that I missed …

Response: We are deeply sorry for that. We have revised the incorrect citation throughout the manuscript. Thanks for the valuable comments.

Line 77. Phage is engineered for enhancing the bactericidal efficacy by

Suggest using plural: Phages are engineered for ….

Response: We agree. We have revised the sentence. Please see line 78.

Line 279

Suggest replacing “sustained” with something like “increased”

Response: We appreciate referee’s comment. We have revised as required. Please see line 282.

Line 287

Amide bond should be amide bond

Response: We agree. We have revised as required. Please see line 291. 

Line 297

  1. Albicans should be C. albicans

Response: Thanks. We have revised as required. Please see line 302.

Line 349 due to their unstable.

Suggest replacing with something like: “due to their limited stability”

Response: We agree. We have revised the sentence. Please see line 355.

Line 362 Therefore, we urged that more clinical trials should be done in the future ….

Suggest replacing with something like:  Therefore, there is an urgent need for more clinical trials to be conducted in the future …

Response: We appreciate referee’s comment. We have revised the sentence. Please see line 368-370.

Line 367 which contained

 Add are. Which are contained

Response: We agree. We have revised. Please see line 373 

Line 270

Ionic binding should be ionic binding (no capital letter for ionic) 

Response: Thanks. We have revised as required. Please see line 273.

Line 377 It needs,

Suggest replacing with something like: There is a need

Response: We appreciate referee’s comment. We have revised. Please see line 384.

Reviewer 3 Report

Dear Authors

The authors improved the manuscript comparing with the previous submission. However, the major point of my comment was repeated in this revised manuscript again. One of the most important aspects of the review paper is transferring the useful information of the relevant topic. And this is supported by the appropriate and adequate use of references. Please remind this.

Sincerely yours

Author Response

Reviewer #3

Comments:

Dear Authors

The authors improved the manuscript comparing with the previous submission. However, the major point of my comment was repeated in this revised manuscript again. One of the most important aspects of the review paper is transferring the useful information of the relevant topic. And this is supported by the appropriate and adequate use of references. Please remind this.

Sincerely yours

Response: We appreciate referee’s comment. We revised the manuscript again according to the referee’s previous comments. In addition, we have our manuscript corrected by a native English speaker. We hope it will be a better version. However, we are still concerned that we may not have accurately captured the meaning of the reviewer on a specific comment, so if some of the changes are still not appropriate, we really hope that the reviewers can give us more detailed comments. We sincerely thank you for your time and effort considering our manuscript. And we will continue to work hard afterwards and try to complete a better paper. Thanks for your valuable comments again.

Line 66: I barely see the “many clinical reports” from the reference. In addition, that research is published at “Experimental Therapeutics” part. I recommend referring the study addressing more broad area, not specific.

Response: We agree. In the new version, we have cited several appropriate references. Please see line 72.

Line 98 (Table1): Authors should provide the correct information for the readers. The mechanism of peptide 1018 is not specifically targeting the ppGpp as stated in the reference 26.

Response: We appreciate referee’s comments. We have made a correction to the mechanism of peptide 1018. Please see line 106 (Table 1).

Line 102: The reference [27] is not appropriate for the sentence. Topic of the study is detection of the antibiotic resistance, not the emerging of MDR genes caused by extensive use of antibiotics.

Response: We agree. We have cited more appropriate reference to support the sentence. Please see line 110.

Line 103-106: Please refer proper references. Very broad statement with very specific and not related references.

Response: We agree. We have revised the sentence and refer proper references. Please see line 110-112.

Line 150: Do authors really think the reference [34, Antimicrobial resistance of Pseudomonas aeruginosa biofilms.] is supporting the bacterial two life style; planktonic and sessile?

Response: We appreciate your comments. We have referred the appropriate reference. Please see line 158.

Line 156-157: In my opinion, they did not observe pathogen biofilm in human tissues in that referred research [37, Dispersing biofilms with engineered enzymatic bacteriophage.]

Response: We appreciate your comments. We have referred the appropriate references. Please see line 164-166.

Line 164-170: “T7 DspB” not “T7DspB”? And provide the proper reference number.

Response: T7 DspB and T7DspB and are identical. We have corrected the reference number. Please see line 176-183.

Line157-158: Please use the supporting reference. This is very specific and I can't find "many” studies.

Response: We agree. We have referred the appropriate references. Please see line 166-168.

Line 171-175: Please provide the reference(s).

Response: We have provided the references. Please see line 184-189.

Line 180-181: Where is the report the authors mentioned?

Response: We have revised the sentence and added reference. Please see line 194-197.

Line 211-212: This sentence should be more clear. There are plenty of phages having broad host range, even crossing the genus. And the main theme of the referred research is the "tunable host range” with the engineering. Again, please provide the correct information.

Response: We cited the reference 62. After carefully reading this reference, we think the appropriate statement may be “the host range of most wild-type phages is limited”, please see line 228-230.

Line 238-250: Reference [57] is a research about database, not for the mechanism of RM system as mentioned in the sentence. And the contents about RM system sounds plausible. However, the referred paper [59] used the "methyltransferase of other bacteria's RM system as it does not interact with the examined bacteria. SEE THE MANUSCRIPT: [We elected to introduce a methyltransferase gene of the type II restriction/modification (R/M) system LlaDCHI (59) from L. lactis since this system is functional in S. thermophilus DGCC7710 and does not interfere with the DGCC7710 CRISPR-Cas systems.] Manuscript should be written based on the findings, not the speculations.

Response: We read the reference in detail and found that we did make a mistake there. Their study used methyltransferase of RM system just for the selection. So the reference is not relevant to our topic. In order to improve the quality of documents, we removed this part.